# The effectiveness of case management interventions for the homeless, vulnerably housed and persons with lived experience: A systematic review

**David Ponka[1], Eric Agbata[2], Claire Kendall[3,4,5], Vicky Stergiopoulos[6], Oreen Mendonca[3], Olivia Magwood[3], Ammar Saad[3,7], Bonnie Larson[8], Annie Huiru Sun[3], Neil Arya[9], Terry Hannigan[3], Kednapa Thavorn[5,7], Anne Andermann[10], Peter Tugwell[11], Kevin Pottie[3,4]***

**1** Department of Family Medicine, University of Ottawa, Ottawa, ON, Canada, **2** Faculty of Health Science, University of Roehampton, London, United Kingdom, **3** C.T. Lamont Primary Health Care Research Centre, Bruyère Research Institute, Ottawa, ON, Canada, **4** Department of Family Medicine and School of Epidemiology and Public Health, University of Ottawa, Ottawa, ON, Canada, **5** Ottawa Hospital Research Institute, Ottawa, ON, Canada, **6** Centre for Addiction and Mental Health, Department of Psychiatry, University of Toronto, Toronto, ON, Canada, **7** School of Epidemiology and Public Health, University of Ottawa, Ottawa, ON, Canada, **8** Department of Family Medicine, University of Calgary, Calgary, AB, Canada, **9** Department of Health Sciences, Wilfred Laurier University, Waterloo, ON, Canada, **10** Department of Family Medicine and Department of Epidemiology, Biostatistics and Occupational Health, McGill University, Montreal, QC, Canada, **11** Faculty of Medicine, University of Ottawa, Ottawa, ON, Canada

* kpottie@uottawa.ca

**Data Availability Statement:** All relevant data are within the paper and its Supporting Information files.

## Abstract

### Background

Individuals who are homeless or vulnerably housed are at an increased risk for mental illness, other morbidities and premature death. Standard case management interventions as well as more intensive models with practitioner support, such as assertive community treatment, critical time interventions, and intensive case management, may improve healthcare navigation and outcomes. However, the definitions of these models as well as the fidelity and adaptations in real world interventions are highly variable. We conducted a systematic review to examine the effectiveness and cost-effectiveness of case management interventions on health and social outcomes for homeless populations.

### Methods and findings

We searched Medline, Embase and 7 other electronic databases for trials on case management or care coordination, from the inception of these databases to July 2019. We sought outcomes on housing stability, mental health, quality of life, substance use, hospitalization, income and employment, and cost-effectiveness. We calculated pooled random effects estimates and assessed the certainty of the evidence using the GRADE approach. Our search identified 13,811 citations; and 56 primary studies met our full inclusion criteria. Standard case management had both limited and short-term effects on substance use and housing outcomes and showed potential to increase hostility and depression. Intensive case

**Funding:** This systematic review was funded by Inner City Health Associates, Toronto, Canada to KP. The funders of the study had no role in the study design, data collection, data analysis, data interpretation, or the writing of the report. The corresponding author had full access to all of the data in the study and had final responsibility for the decision to submit for publication.

**Competing interests:** David Ponka, Claire Kendall, Vicky Stergiopoulos, Anne Andermann, Peter Tugwell and Kevin Pottie are principal investigators in an ongoing project to develop Canadian evidence-based guidelines for providing social programs and healthcare services to people who are homeless and vulnerably housed. Terry Hannigan was paid an honorarium by the Bruyère Research Institute to provide consultations on this work. This does not alter our adherence to PLOS ONE policies on sharing data and materials. The authors declare no other conflicts of interest.

management substantially reduced the number of days spent homeless (SMD -0.22 95% CI -0.40 to -0.03), as well as substance and alcohol use. Critical time interventions and assertive community treatment were found to have a protective effect in terms of rehospitalizations and a promising effect on housing stability. Assertive community treatment was found to be cost-effective compared to standard case management.

## Conclusions

Case management approaches were found to improve some if not all of the health and social outcomes that were examined in this study. The important factors were likely delivery intensity, the number and type of caseloads, hospital versus community programs and varying levels of participant needs. More research is needed to fully understand how to continue to obtain the increased benefits inherent in intensive case management, even in community settings where feasibility considerations lead to larger caseloads and less-intensive follow-up.

## Introduction

Homeless and vulnerably housed populations have poorer health outcomes including acute and chronic illness [1], traumatic injury [1], mental health and substance use disorders [2–7], and mortality [8]. While often related to individual medical and complex social needs, structural challenges posed by fragmented health and social systems create a potent mix of barriers to access to health care. These include a lack of sufficient language capacity, awareness of affordable healthcare services and their location, transportation services, childcare, and reasonable wait times. When coupled with previous experiences of rejection or discrimination from service providers, these barriers further contribute to individuals failing to access appropriate and available health care [9–11].

To address these barriers, people who are homeless or vulnerably housed may benefit from tailored, patient-centered care with an integrated approach to community and social services [12–14]. Case management (CM) is one such intervention where individual case managers respond to the complexity of navigating the healthcare system by assessing, planning and facilitating access to health and social services [15,16]. While case management interventions are heterogeneous in definition, complexity, target populations served, and modes of delivery [12], among these, four predominant models have evolved in relation to health care: standard case management (SCM), intensive case management (ICM), assertive community treatment (ACT), and critical time intervention (CTI) (See Table 1) [17].

Case management has been shown to improve patient satisfaction [27], quality of life, and the utilization of community-based services among other high-risk populations [28]. However, the evidence base for CM and its implementation among homeless and vulnerably housed populations remains sparse. This review is one of a series of reviews on the effectiveness of providing interventions for homeless and/or vulnerably housed persons. The objective of this review is to assess the effectiveness and cost-effectiveness of four CM models for the health and social outcomes of homeless or vulnerably housed individuals in the following domains: housing stability, mental health, substance use, quality of life, hospitalization, employment and income.

**Table 1. Characteristics of case management models- Adapted from de Vet et al. 2013 [15].**

| | Standard Case Management | Intensive Case Management | Assertive Community Treatment | Critical Time Intervention |
|---|---|---|---|---|
| Focus of Services | Coordination of services | Comprehensive approach addressing several needs (i.e. housing, physical and mental health, addictions services etc.) | Comprehensive approach addressing several needs (i.e. housing, physical and mental health, addictions services etc.) | Targeted to continuity of care between a period of transition i.e. between precarious housing conditions (i.e. living in a shelter or discharged from hospital) and independent housing arrangements |
| Target Population | Homeless persons with complex health concerns | Homeless persons with the greatest service need i.e. persons with serious mental illnesses, but typically fewer hospitalizations or less functional impairments [18], and for people experiencing addictions [19]. | Homeless persons with the greatest service need i.e. for persons with serious mental illness, often schizophrenia or bipolar disorder, accompanied by a history of multiple psychiatric hospitalizations and functional impairment [20]. | Homeless persons at critical transitions in their lives i.e. between a shelter or hospital and independent housing |
| Access Point | Varies by location. Typically services are accessed through a referral by healthcare professionals (clinician, nurse, social worker, outreach worker). Some locations offer self-referral services where clients can apply for access to services on their own [21]. | | | |
| Duration of Services | Time limited. once the case manager has brokered the client to a service provider, the service provider to provide ongoing support until a positive outcome is achieved [15]. | Ongoing | Ongoing but transfer to lower intensity services is common after a period of stability [22,23]. | Time-limited. Usually a period of 9 months after institutional discharge or placement in housing [22]. |
| Availability of case management services | | up to 12 hours per day, 7 days a week [24]. | 24 hours per day, 7 days per week availability [22]. | |
| Where services are offered | Brokering of services to other providers [25]. | Case manager accompanies clients to meetings and appointments [24]. | Services are offered in a natural setting such as the workplace, home or social setting [15,22]. | Worker provides services in the home and helps to strengthen community networks [22]. |
| **Coordination** of access to services run by other agencies or **service provision** by the agency itself | Coordination | Coordination and service provision | Coordination and service provision | Coordination and service provision |
| Average Caseload (program intensity) | 35 | 15 | 15 | 25 |
| Outreach | No | Yes | Yes | Yes |
| Responsibility for clients' care | Case managers can originate from several different teams (a mental health team, addictions care team, primary care health team, shelter team, Housing First etc.). Regardless of the team, all case managers play the role of navigator and keep the client's needs at the forefront of their care. | | | |
| | Case manager or a navigator role is played by a clinician, nurse, community outreach worker, or social worker [15,26]. | Case manager | A multidisciplinary team including case managers, peer support workers, and physicians [20]. | Case manager or CTI worker [22]. |
| Case example | Client is homeless or vulnerably housed with no serious mental illness or addictions concerns. Client accesses SCM. Here a clinician, nurse, social worker or outreach worker to play the role of a standard case manager and refer to needed services. | Client is homeless or vulnerably housed with a serious mental illness and/or addiction concern. Client accesses ICM. Here a case manager will arrange for needed assistance and will accompany them to services. | Client is homeless or vulnerably housed with a serious mental illness and/or addiction concern and a history of recurrent hospitalizations. Client accesses ACT. A multidisciplinary team led by a case manager, will offer services in the client's natural setting (home/workplace). | Client is homeless or vulnerably housed and is in a period of transition (i.e. from a shelter or hospital into a housing unit). Client accesses CTI where a case manager or CTI worker will broker or provide services to help with the transition. |

## Methods

### Protocol registration and reporting

We conducted a systematic review according to a published peer-reviewed protocol [29]. The protocol was not registered in an open-access registry (e.g. PROSPERO) prior to publication.

We followed the PRISMA checklist and SWiM (Synthesis Without Meta-Analysis) reporting guidelines when reporting our findings (see S1 File) [30,31]. Ethical approval was not required for this study.

## Selection of priority interventions

We conducted a Delphi consensus process with 84 experienced healthcare practitioners and 76 persons with lived homelessness experience to prioritize person-centered and clinically meaningful priority topics, outcomes, and subgroups [32]. Among these, case management and care coordination were highly prioritized. We then scoped literature using Google Scholar and PubMed to broadly determine a list of interventions and terms relating to each of the Delphi priority topic categories. A working group was formed to arrive at a consensus and inform the final selection of interventions to be included in this review. This working group consisted of medical practitioners, allied health professionals, and community scholars (people with lived experience of homelessness or vulnerable housing) [33]. Our working group deliberated the value of systematic reviews and evidence-based guidelines on various interventions, giving significant weight to the needs and opinions of persons with lived experience of homelessness. Consensus of the working group was to describe case management interventions by level of intensity (Table 1)

## Search strategy and selection criteria

A search strategy was developed and peer-reviewed by a health science librarian. We searched MEDLINE, Embase, CINAHL, PsycINFO, Epistemonikos, HTA database, NHSEED, DARE, and the Cochrane Central Register of Controlled Trials (CENTRAL) from the inception of these databases to February 8, 2018, for studies on effectiveness, cost and cost-effectiveness. A combination of indexed terms, free text words, and MeSH headings were used (See S2 File). There were no date or language restrictions. We searched the reference lists of relevant systematic reviews for studies that met our inclusion criteria. We consulted experts in the field of homelessness and people with lived experience to identify any additional studies we may have missed. We updated our search on July 19, 2019 and deduplicated against our previous search to identify trials published since February 2018.

The results were uploaded to Rayyan reference manager software to facilitate the study selection process [34]. Teams of review authors assessed each study for inclusion in duplicate (See Table 2); disagreements were resolved through discussion or a third reviewer. All peer-reviewed studies that assessed case management interventions among homeless or vulnerably housed populations and that reported on relevant outcomes were included. We excluded articles where case management was delivered as a component of a permanent supportive housing intervention as this is covered by a parallel review [35].

## Data analysis

We used a standardized data extraction sheet that included the study methodology, population, intervention, control, outcome, study limitations, and funding details. The data were extracted independently by two reviewers. Disagreements were resolved through discussion. To prevent double-counting of outcomes, individual records were carefully screened to identify unique trial studies. Each study was then evaluated for potential overlap using study design, enrollment and data collection dates, authors and their associated affiliations and the reported selection and eligibility criteria in the studies to inform the assessment. Studies deemed to be at risk for double-counting were discussed by the research team and decisions for inclusion in meta-analysis (and any additional analyses) were made. We used the Cochrane Risk-of-Bias tool to assess the quality of each study's methodology, in duplicate [40].

**Table 2. Eligibility criteria.**

| Study Characteristics | Inclusion Criteria | Definitions |
|---|---|---|
| Population | People experiencing homelessness and vulnerable housing. If study populations were heterogeneous, we included the study if the population was comprised of >50% homeless or vulnerably housed individuals. | |
| Interventions | Standard Case Management (SCM) | These allow for the provision of an array of social, healthcare, and other services with the goal of helping the client maintain good health and social relationships. This is done by "including engagement of the patient, assessment, planning, linkage with resources, consultation with families, collaboration with psychiatrists, patient psychoeducation, and crisis intervention" [36]. |
| | Intensive Case Management (ICM) | ICM helps service users maintain housing and achieve a better quality of life through the support of a case manager that brokers access to an array of services. The case manager accompanies the service user to meetings and can be available for up to 12 hours per day, 7 days a week. Case managers for ICM often have a caseload of 15–20 service users each [15]. |
| | Assertive Community Treatment (ACT) | ACT offers team-based care by a multidisciplinary group of healthcare workers in the community. This team has 24 hours per day, 7 days per week availability and provides services tailored to the needs and goals of each service user [15,23]. |
| | Critical Time Intervention (CTI) | CTI is a service that supports continuity of care for service users during times of transition; for example, from a shelter to independent housing or following discharge from a hospital. This service strengthens the person's network of support in the community [37]. It is administered by a CTI worker and is a time-limited service, of usually a period of 6–9 months. |
| Comparison | No intervention, standard intervention, alternative intervention, treatment as usual. | |
| Outcomes | Housing stability, mental health, quality of life, substance use, hospitalization, income, and employment-related outcomes. | |
| Study Characteristics | Primary studies as defined by EPOC criteria [38] Randomized controlled trials Non-randomized controlled trials Controlled before-after studies Interrupted time series and repeated measures studies Cost or cost-consequence studies Full economic evaluation studies: cost-minimization analysis, cost-benefit analysis, cost-effectiveness analysis, and cost-utility analysis. All study designs must include interventions with a comparison/control group and have measured outcomes. | |
| Study Characteristics | **Exclusion Criteria** | **Justifications** |
| | Studies taking place in low- middle-income countries [39]. | Due to the variability in access to resources and supports in comparison to that in a high-income country vary greatly. We feel that the settings are different and should be synthesized separately |
| | Studies that exclusively report on Indigenous specific interventions | The analysis of the interventions tailored to this population will be covered by an Indigenous research group. |
| | Case management delivered as a component of a permanent supportive housing intervention | This is covered by a parallel systematic review [35]. |

Where possible, we conducted meta-analysis of measures of effectiveness using random effects models due to their consideration of heterogeneity using RevMan 5.3 software [41]. We verified that the random effects model did not under-estimate the confidence intervals by running parallel fixed effects analyses. We present the summary effects as relative risks or

**Table 3. GRADE certainty of evidence and definitions.**

| Certainty rating | Definition |
|---|---|
| High | Further research is very unlikely to change our confidence in the estimate of the effect |
| Moderate | Further research is likely to have an important impact on our confidence in the estimate of the effect and may change the estimate |
| Low | Further research is very likely to have an important impact on our confidence in the estimate of the effect and is likely to change the estimate |
| Very low | Any estimate of the effect is very uncertain |

Source: [43]

standardized mean differences, as appropriate. Where study heterogeneity did not allow for meta-analysis, we employed a narrative synthesis, defined as a "synthesis of findings from multiple studies that relies primarily on the use of words and text to summarise and explain the findings of the synthesis. Whilst it can involve the manipulation of statistical data, the defining characteristic is that it adopts a textual approach to the process of synthesis to 'tell the story' of the findings from the included studies" [42]. We used the GRADE approach to appraise the certainty of the evidence (See Table 3) [43].

## Results

We identified 11,934 citations from bibliographic databases and an additional 17 from other sources. After removing duplicates, we screened 7,514 titles and abstracts for eligibility. We assessed 268 citations at full-text, of which 214 were excluded (See Fig 1 and S3 File). Our updated search yielded a total of 1877 additional records, of which 1869 records were screened by title and abstract after removing duplicates. We assessed 36 articles at full text, of which 34 were excluded (See Fig 2). From both searches, we included a total of 56 citations, of which 11 reported on SCM [44–54], 10 on ACT [25,55–63], 17 on ICM [64–80], and 11 on CTI [81–91]. Twelve articles provided evidence on cost-effectiveness; 3 on SCM [50,79,92]; 6 on ACT [56,59,93–96]; 2 on ICM [97,98]; and 1 for CTI [89] (See Figs 1 and 2). Five of the cost-effectiveness articles were included in the effectiveness analysis as well [50,56,59,79,89]. The majority of the included studies were set in the United States, with three studies from Europe and one from Australia. All of the studies focused on homeless and vulnerably housed populations, with varying levels of participant profiles and comorbidities across studies. All trials compared case management interventions to usual care (UC) or an alternative intervention, such as rent vouchers, peer support groups or drop-in services. Appendix S4 lists the characteristics of the included studies on SCM, ICM, ACT, CTI and cost-effectiveness studies.

### Characteristics of included studies (SCM)

The effects of all of the case management interventions are summarized in Table 4. In our risk-of-bias assessment (See S5 File), we found that the majority of studies had methodological deficiencies in randomization, allocation concealment and blinding of participants and personnel. The GRADE certainty of the evidence for critical patient-important outcomes is available in S6 File.

### Effects of standard case management (SCM)

Of 11 trials on SCM, ten evaluated housing stability [44–48,50–54]. Only three reported significant decreases in homelessness [44,51,52]; an effect that diminished over time in one trial of a time-limited residential case management where participants in all groups accessed significant levels of services [44].

A SCM program tailored to women reduced the odds of depression at 3 months (OR 0.38 95% CI 0.14 to 0.99) but did not show improvements in their overall mental health status (MD 4.50; 95% CI -0.98 to 9.98) [53]. One trial reported *higher* levels of hostility (p<0.001) and depression symptoms (p<0.05) among female participants receiving nurse-led SCM compared to those receiving standard care, although no significant difference in psychological well-being was reported between these groups [49]. Two additional trials reported no impact on mental health outcomes [44,54]. Two trials reported decreased problematic substance use [44,79], and four others reported no effect on this outcome [48–50,53].

Findings were equivocal for quality of life outcomes. One trial compared health advocate SCM (with or without outreach registration) to usual care [45,46]. While some quality of life

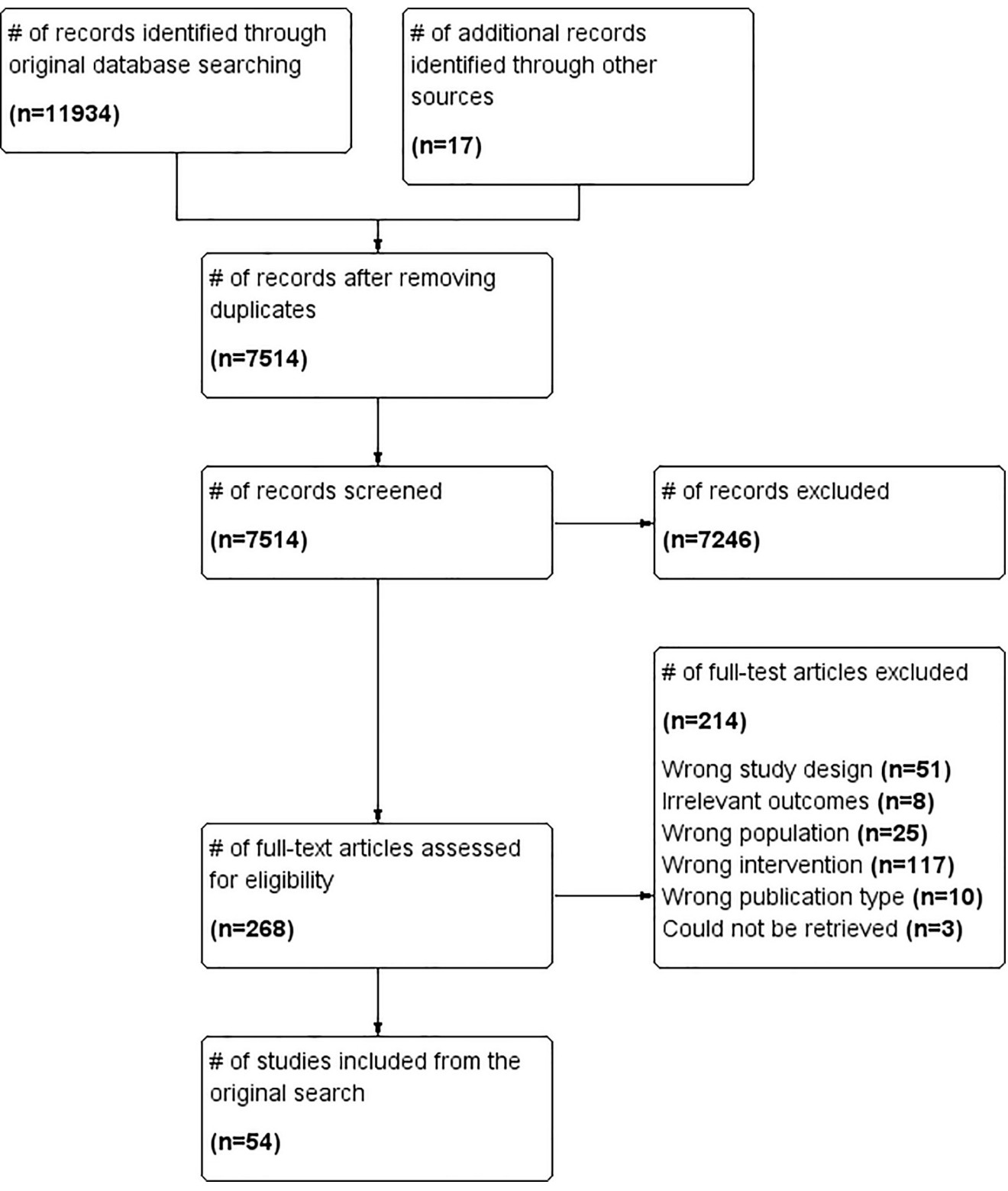

**Fig 1. PRISMA flow diagram of search up to February 2018.**

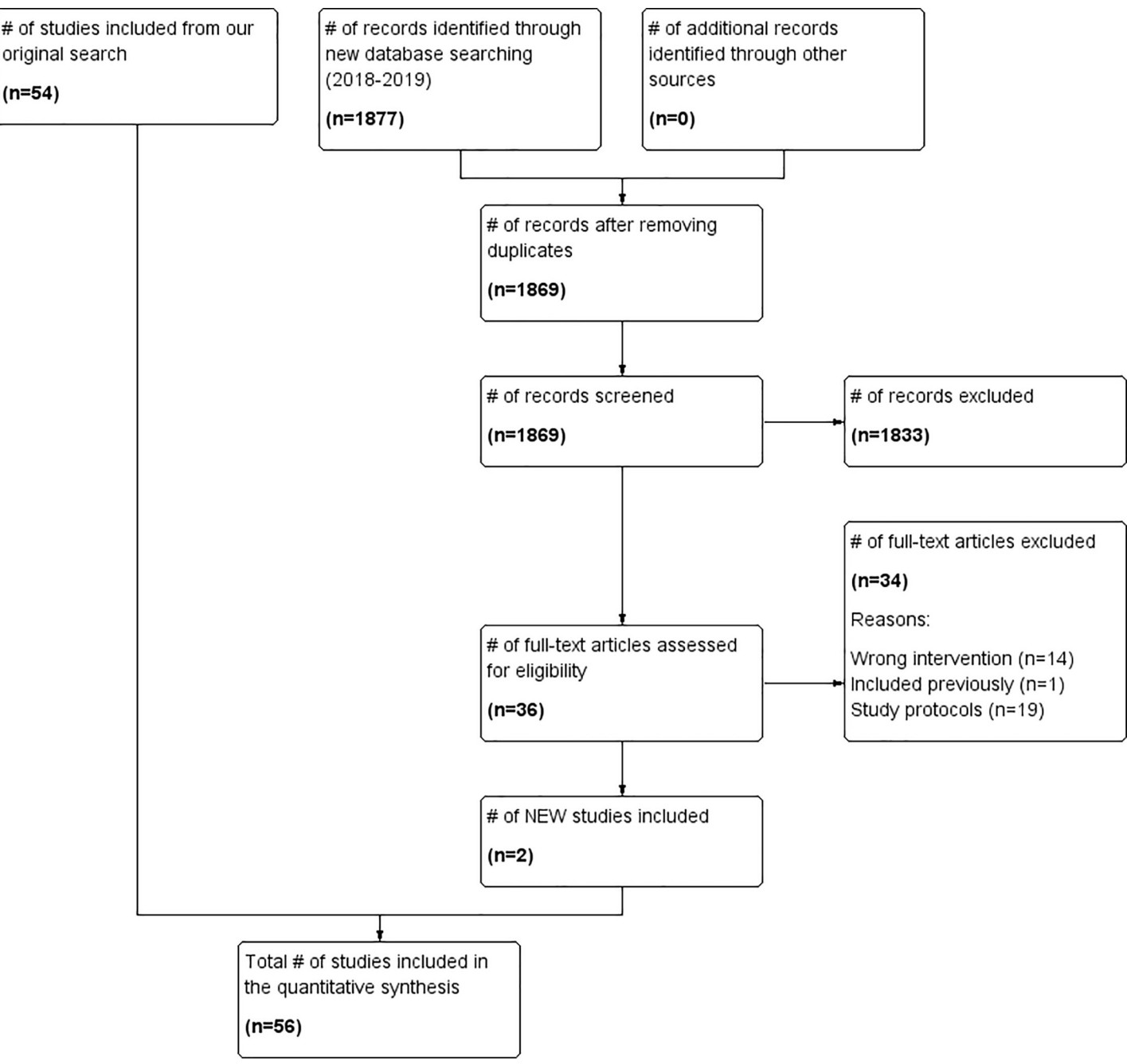

**Fig 2. PRISMA flow diagram with updated search up to July 2019.**

domains (e.g. social isolation, sleep) favored health advocate SCM, most effects on quality of life were not significant. Another trial reported no significant benefits of nurse-led SCM on life satisfaction scores [49].

A single trial of health advocate SCM (with or without outreach registration) assessed health service utilization over three months [46]. Only five percent of all participants accessed the emergency department, with no significant difference between health advocacy or usual care groups [46]. Finally, five studies assessed the effectiveness of SCM on employment

**Table 4. Results of studies comparing assertive community treatment, intensive case management, critical time interventions, and standard case management to control services.**

| Intervention* | Study ID | Is the between-group difference significantly favouring the case management intervention? | | | | | | |
|---|---|---|---|---|---|---|---|---|
| | | Housing stability | Mental health | Quality of life | Substance use | Hospitalization | Employment | Income |
| ACT | [55] | No | - | - | - | No | - | - |
| ACT | [56] | Yes | No | Yes [1,3] | - | Yes [1,3] | - | No |
| ACT | [25] | - | No | No | Yes [2] | Yes [2] | - | - |
| ACT | [57] | Yes [2] | No | - | No | - | - | - |
| ACT | [58] | Yes [1] | No | Yes [1] | - | Yes | - | - |
| ACT | [61] | Yes [2] | No | - | No | - | - | No |
| ACT | [62,63] | Yes [2] | Yes [1] | - | No | - | - | No |
| ACT | [59,60] | No | No | - | No | - | - | - |
| ICM | [64] | No | No | No | Yes [1] | - | No | - |
| ICM | [65] | No | No | - | No | - | - | - |
| ICM | [66] | - | Yes [1] | Yes [1] | No | - | - | - |
| ICM | [67] | Yes [2] | - | - | - | - | - | - |
| ICM | [68,69] | Yes | - | - | Yes [1] | - | No | Yes [1,3] |
| ICM | [70] | - | No | Yes [1,2] | - | - | - | - |
| ICM | [71] | No | - | - | - | - | - | No |
| ICM | [72] | Yes | - | - | - | No | - | - |
| ICM | [73] | No | No | - | Yes [1] | No | - | - |
| ICM | [74] | No | No | No | - | No | No | - |
| ICM | [75] | Yes [3] | Yes [3] | - | Yes [3] | - | Yes [3] | - |
| ICM | [76] | No | - | - | No | Yes | - | Yes |
| ICM | [77] | Yes [1] | Yes [1] | Yes [1] | - | - | - | - |
| ICM | [78] | Yes [2] | No | - | Yes [2] | - | No | - |
| ICM | [79] | Yes | No | - | Yes | Yes [2] | - | Yes |
| ICM | [80] | No | Yes [1] | - | No | - | - | No |
| CTI | [81] | No | No | No | No | - | - | - |
| CTI | [82,83] | Yes [1,3] | - | - | - | Yes [1,3] | - | - |
| CTI | [84] | - | Yes [1] | No | | | | |
| CTI | [85,86] | Yes [1] | Yes [1,2] | - | - | - | - | - |
| CTI | [87–91] | Yes [1] | - | - | - | No | - | No |
| SCM | [44] | Yes [diminished with time] | No | - | Yes [1] [diminished with time] | - | Yes | - |
| SCM | [45,46] | No | - | Yes [1] | - | No | - | - |
| SCM | [47] | No | - | - | - | - | - | - |
| SCM | [48] | No | - | - | No | - | No | - |
| SCM | [49] | - | HARMS | No | No | - | - | - |
| SCM | [50] | No | - | - | No | - | No | - |
| SCM | [51] | Yes | - | - | Yes | - | No | - |
| SCM | [52] | Yes | - | - | - | - | - | - |
| SCM | [53] | No | Yes [1] | - | No | - | - | - |
| SCM | [54] | No | No | - | - | - | No | - |

*Assertive Community Treatment; ACT. Intensive Case Management; ICM. Critical Time Intervention; CTI. Standard Case Management; SCM.

**1**. Depends on sub-outcomes

**2**. Depends on sub-groups

**3**. Depends on analysis methodology

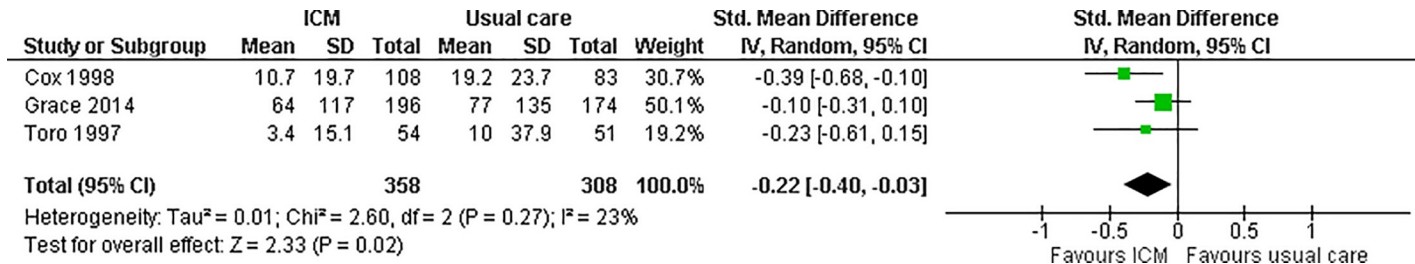

**Fig 3. ICM versus usual care pooled analysis of number of days spent homeless (long term, 13+ months follow-up).**

outcomes. One trial reported a significant improvement in employment over 24 months [44], whereas four trials showed no significant difference [48,50,51,54]. While one trial suggests that SCM improves access to income assistance (p<0.05) [51], no trials on SCM measured participant income as an outcome.

### Effects of intensive case management (ICM)

Fourteen of sixteen trials on ICM assessed housing stability [64,65,67,68,71–80]. Overall, ICM showed small positive effects on housing outcomes, with seven of these fourteen studies [67,68,72,75,77–79] suggesting improvements in housing stability and the other seven reporting no effect (Table 4). A pooled analysis shows that ICM significantly reduced the number of days spent homeless (SMD -0.22 95% CI -0.40 to -0.03; See Fig 3) but had no significant effect on the number of days spent in stable housing compared to usual services (See Fig 4). These findings were unchanged regardless of whether random effects or fixed effects models were used in the analysis (See S7 File). For time-limited interventions, ICM effectively housed more participants [72], reduced time spent in community housing, streets and shelters [77], and reduced the number of moves to different residences [71]. Three other trials reported that ICM was associated with no difference on the number of days in no-rent or privately rented accommodations, better or worse accommodations, stable housing or homelessness compared to standard case management or usual services [74,75,78].

ICM had mixed effects on mental health outcomes. Four trials reported significant reductions in psychological symptoms [66,75,77,80], whereas seven additional trials reported no effect [64,65,70,73,74,78,79]. In two trials, positive mental health outcomes were correlated with improvements in quality of life [66,77], with an additional trial reporting better quality of life despite no significant differences in mental health [70]. Only one trial reported no effect of ICM on quality of life [74].

ICM had a significant benefit in reducing substance use in six of ten trials that measured this outcome [64,68,73,75,78,79]. ICM was associated with significant reductions in alcohol consumption [68,73,75] and reductions in problematic drug use [64,78,79].

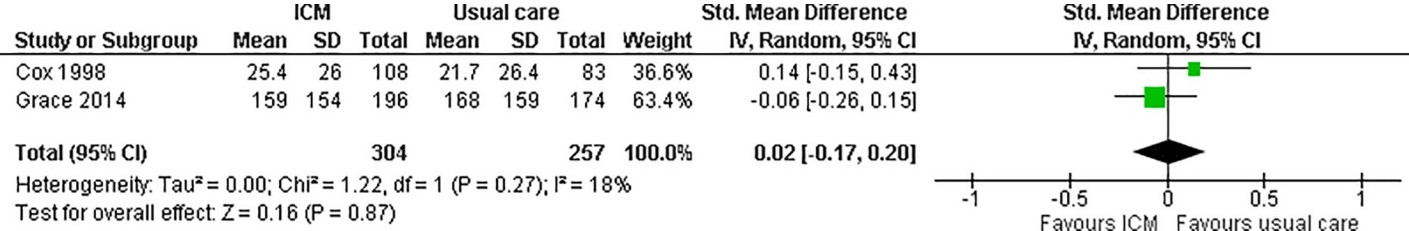

**Fig 4. ICM versus usual care pooled analysis of number of days spent in stable housing (long term, 13+ months follow-up).**

ICM had mixed effects on participants' hospitalization outcomes. Two studies reported significant reductions in the number of emergency department visits but not in the use of other hospital services compared to usual care [76,79]; while three additional trials reported no significant reductions in the number of days in hospital compared to usual services or support groups [72–74].

Finally, the effect of ICM on income and employment outcomes was small. In one study, ICM was associated with increased number of days paid from employment [75], which was not found in four other trials [64,68,74,78]. Three studies reported that ICM was significantly associated with increased attainment of public income assistance and reduced the incidence of unmet financial need [79] among single adults [68,76]. However, among youth [71], and families [80], ICM had no impact on income obtained from employment or public assistance.

## Effects of assertive community treatment (ACT)

Assertive community treatment showed promising effects on housing stability in five of seven trials that measured this outcome [56–58,61,62]. Participants who received ACT reported significantly more days in community housing (p = 0.006) [58], and fewer days homeless (p<0.01) compared to usual or supportive services [61]. ACT marginally improved the number of days participants spent in stable housing compared to supportive services (p = 0.032) [62], and usual services (p = 0.09) [57]. However, two trials, one of which included a follow-up study, did not identify any housing-stability benefits of ACT over usual or supportive services [55,59,60].

The effects of ACT on mental health outcomes were moderately positive. In one trial, ACT interventions were associated with fewer psychological symptoms in the areas of unusual activity levels (p<0.03) and thought disorder (p<0.02) compared to other supportive services [62]. Six other trials reported no additional effects of ACT on mental health compared to usual or supportive services [25,56–59,61]. ACT had equivocal effects on substance use outcomes. One trial showed that ACT participants with more severe alcohol use disorder experienced faster and earlier improvements in substance use compared to those with less severe alcohol-use disorder or those randomized to usual or supportive services (p<0.01) [25]; however, this difference was not significant by the end of three years. Four trials reported no additional benefits of ACT on substance use outcomes over usual or supportive services [57,59,61,62].

Findings on quality of life outcomes were mixed. One trial reported that ACT was significantly associated with better overall quality of life over 18 months compared to those receiving SCM (p<0.05) [56]. Another trial found no significant improvements for ACT over usual care in objective quality of life measures over 12 months, although ACT participants showed earlier improvement in life satisfaction rates compared to usual care at 6 months (p = 0.005) [58]. A third trial found no additional effects of ACT on quality of life outcomes compared to usual and supportive services [25].

Findings on hospitalization outcomes were mostly positive. One trial reported that ACT participants spent approximately half as many days in the hospital compared to those receiving standard case management [56]. No significant differences between groups were found on time to discharge from hospital or length of hospitalization. Another trial showed that ACT was associated with significantly fewer days hospitalized over 3 years compared to other supportive services (MD 19; p = 0.002) [25]. One trial reported fewer emergency department visits for ACT participants compared to usual care at 12 months (p = 0.009) [58], whereas another trial found no effect of ACT over usual care on either days in hospital or emergency department visits [55].

Finally, three trials reported no effect of ACT on income outcomes over usual or supportive services [56,61,62]. No trials measured employment outcomes.

## Effects of critical time interventions (CTI)

Critical time interventions showed a promising effect on housing stability in three of four trials [82,85,87]. In the US context, one trial found that CTI significantly reduced the number of days spent homeless during the final 18 weeks of the study, compared to usual services (OR 0.22; 95% CI 0.06 to 0.88) [82]; however, this effect was not significant over the entire 18 months of the trial. Another trial reported a significant reduction in the average number of nights spent homeless among CTI participants compared to usual services over 18 months (Difference = -61; p = 0.003) [87]. Families that received CTI transitioned from shelter to housing more rapidly than the usual services group (MD -107.9 days; 95% CI -136.2,-79.6) [86]. Conversely, one European trial found that CTI did not have any impact on days rehoused after a 9-month period compared to usual services [81].

CTI showed little effect on mental health outcomes. However, a trial conducted among abused women reported significantly fewer symptoms of PTSD during follow-up (Adjusted MD -7.27, 95% CI -14.31 to -0.22, p = 0.04), but no effect on symptoms of depression or psychological distress [84]. In another RCT [85], families who received CTI showed mixed results on the frequency of children's internalizing and externalizing problems.

Two RCTs examined quality of life outcomes and found no significant impact of CTI over usual services at 9 months [81,84]. As well, when looking at substance-use outcomes, CTI was associated with non-significant reductions in cannabis and alcohol use [81].

One study found that CTI was significantly associated with reduced odds of rehospitalization (OR 0.11, 95% CI 0.01 to 0.96, p = 0.07) and total number of nights hospitalized (p<0.05) in the final 18 weeks of the trial [83]. Another trial suggests that CTI reduced the total number of nights of hospitalization over 18 months but not the average length of hospital stays [88].

Finally, one trial showed no significant effect of CTI on income-related outcomes compared to usual services [89]. No trials reported on employment-related outcomes.

## Cost and cost-effectiveness of the interventions

Evidence on cost and cost-effectiveness was mixed. The total cost incurred by SCM clients was higher than those receiving usual or standard care [50,79], but lower compared to a US clinical case management program that included housing vouchers and ICM [98]. Cost-effectiveness studies showed that when the benefits gained and costs borne to all payers were considered (also known as a societal perspective) SCM was not cost-effective compared to ACT for persons with serious mental disorders or those with a concurrent substance-use disorder as it was both more expensive [56,94], and was associated with more days in unstable housing [56], and poorer quality of life [94]. SCM was slightly more costly than ACT because SCM clients had nominally more frequent visits to outpatient health care and other community services, more arrest episodes, and incurred higher family time costs compared to ACT clients. For ICM, Stergiopoulos and colleagues showed that the cost of supporting housing with ICM could be partially offset by reductions in the use of emergency shelters and in single-room occupancies [97]. ICM was reported as likely to be cost-effective when all costs and benefits to society are considered [98]. A pre-post study found that when ICM was provided to high users of emergency departments there was a net hospital cost savings of USD$132,726 [92]. For ACT, the included studies that focused on individuals with severe mental illness or dual disorders consistently showed that ACT interventions were associated with lower costs and improved health outcomes compared to the outcomes of usual care [56,59,94–96]. We identified only one study on the cost-effectiveness of CTI which reported that the CTI provided to men with severe mental illness had comparable costs (US$52,574 vs. US$51,749) despite fewer nights spent homeless (508 vs. 450 nights) compared to usual services [89].

## Discussion

We conducted a comprehensive systematic review of four case management interventions for people who are homeless or vulnerably housed. The interventions were complex, and the study populations, intervention intensity, and outcomes were heterogeneous, making it challenging to generalize our findings. However, we can make some overarching statements to guide policy and practice. In general, standard case management showed little to no benefit across any of our outcome domains and in one trial [49], implementing SCM was associated with elevated levels of hostility and depression. We found that interventions of greater intensity, such as intensive case management, assertive community treatment and critical time intervention, did improve several outcomes of interest, most notably housing stability. ICM was found to reduce substance use in several studies and CTI to marginally reduce psychological symptoms; however, there was little impact on the quality of life across studies. ICM was associated with a reduced number of emergency department visits but not of hospital admissions, and both ACT and CTI, overall, showed significant reductions in both the number of emergency department visits and days in hospital. Only ICM was found to consistently improve income outcomes, with significant improvements in access to financial assistance and reductions in unmet financial needs. Case management interventions, especially ACT, were cost-effective for persons with complex needs, including those with severe mental illness or dual disorders, if the overall costs and benefits to patients, health care systems and society as a whole were considered.

Our findings suggest that the effectiveness of case management interventions is related both to the intensity of models as well as to their ability to address and advocate for the comprehensive needs of specific groups such as those with severe mental health conditions or those experiencing transitions in care. Findings suggested that the case management needed to be continuous, community-based and intensive so as to maintain and/or increase the gains achieved. For example, in Sosin and colleague's trial [51], improvements in housing stability were attributed to the case worker's advocacy for access to income benefits and help with locating housing. Not surprisingly, higher intensity case management models, which generally have lower caseloads, also include the provision of services above and beyond care coordination and incorporate outreach services, especially in the case of ICM, which is shown to have greater effects compared to other less intensive case-management models. This may be due to their capacity to address some of the underlying social determinants of health that contribute to the cycle of homelessness, such as poverty, which requires longitudinal engagement with case managers. A parallel review also suggests that case management can have significant impacts when provided in conjunction with permanent housing [35]. Given the heterogeneity of these complex interventions, we cannot be certain of the precise mechanisms and key features that promote effectiveness. However, it is likely that a dose-response relationship may explain some of our findings, and that as higher intensity interventions such as ACT and ICM are more precisely defined, there may be greater attention to fidelity in their implementation [19]. Alternatively, it is possible that lower intensity models work predominantly for homeless populations with less acute issues (or for those that are precariously housed), and this would suggest the importance of matching the intensity of the intervention with the acuity of need. Some indicators from a parallel qualitative review point to a case-manager-client relationship built on trust and continuity of care and integrated services as being key factors in the success of case management programs [99]. Many programs include peers and people with lived experience acting in case management roles [100–103], and while this has been identified as important to those confronted with homelessness [104–106], such approaches require formal evaluation.

These findings contribute to an expanding evidence base on effective interventions for people who are homeless or vulnerably housed. Our review builds on a previous review by De Vet [15] as it incorporates evidence up to 2019 and also considers a broader definition of standard case management that includes health advocates, as well as residential and disease-specific case management. Our study includes studies from the US, Europe and Australia, allowing us to make inferences about more diverse health and social systems which are important to address homelessness as an international public health priority [15]. Overall, our findings are congruent with De Vet's conclusions, but with some important differences. Notably, we saw fewer significant results in access to housing among recipients of CTI, likely arising from differences in healthcare and social contexts. The intensity of "usual care" in the Netherlands was high compared to the US context, where follow-up services were not typically available. Additionally, the Netherlands has an extensive social housing system; thus, reducing the short-term risk of recurrent homelessness. More recent CTI studies also suggest lower rates of rehospitalization than was found in our review. Finally, our broader inclusion criteria of SCM interventions allowed us to identify potential harms, such as higher levels of hostility and depression among case management recipients. Overall, our findings are in agreement with other earlier reviews, including those of Coldwell and Bender [23], Hwang [107], Vanderplasschen [28], and Mueser [108]. We also incorporated cost-effectiveness, and while the results were mixed, they provide important evidence on the potential economic impact of case management interventions on health care systems and society.

In the studies reviewed, the quantitative synthesis was complicated by the heterogeneity that exists across interventions. In addition, there is a lack of clarity in and overlap of the nomenclature used to define different case management interventions [12]. Furthermore, few studies provided the level of intervention detail required to make concrete recommendations with respect to the types of activities conducted, the roles and responsibilities of the case managers, and the postulated mechanisms of success that could inform future practice. Such lack of detail can further contribute to challenges in implementation and fidelity across interventions.

To our knowledge, this is the first systematic review to consider a broad range of outcomes and cost-effectiveness of these types of case-management interventions. We used high quality methods to synthesize randomized controlled trials and controlled trials, conducted meta-analyses, and used GRADE methods to assess the certainty of the effects. We integrated persons with lived experience of homelessness into our research team to ensure the relevancy of this work. Limitations include heterogeneous interventions and populations that precluded quantitative synthesis; thus, the studies were too few to allow us to conduct meta-analyses for the many included outcomes. As the majority of studies were conducted in the United States, our findings may not be generalizable to contexts with substantially different health and social systems. Poorly defined control or "usual care" groups further complicates the relative effectiveness of one case management model over another—a particular issue for SCM models. A weakness inherent to a secondary analysis is the potential for bias with respect to the reporting of results for multiple outcomes. Further, we restricted our inclusion criteria to rigorous experimental study designs, thereby, excluding observational studies that may have provided additional evidence in this area. This review is quantitative in nature and we may have excluded important findings related to case management found in the qualitative literature.

In summary, helping people who are homeless and vulnerably housed navigate and access a complex system of services yields positive outcomes in areas such as housing stability and mental health. Case management interventions may be most effective when they target specific complex populations or times of transition with more effective interventions that involve low caseloads, greater intensity and continuity of contact time, and direct service provision in

addition to mere coordination. More research is needed on SCM models and their ideal target populations. Further, there is a need to more formally evaluate how to best integrate case management into delivery models such as chronic care management programs [109–111], and patient medical home approaches [112,113]. We postulate that further work is required to understand how to embed such interventions in the primary care setting, given the appeal of its continuous and comprehensive nature [114,115]. We suggest future research should apply a realist lens in order to further understand the critical elements and implementation strategies of case management interventions [116,117].

## Supporting information

**S1 File. PRISMA checklist.**
(PDF)

**S2 File. Search strategy.**
(PDF)

**S3 File. List of excluded studies.**
(PDF)

**S4 File. Characteristics of included studies.**
(PDF)

**S5 File. Risk of bias summary.**
(PDF)

**S6 File. GRADE evidence profiles.**
(PDF)

**S7 File. Fixed and random effects analyses.**
(PDF)

## Acknowledgments

The authors would like to acknowledge Doug Salzwedel for the systematic search, as well as the following working group members for technical support in screening, data extraction and/ or critical appraisal: Tasnim Abdalla, Michaela Beder, German Chique-Alfonzo, Wahab Daghmach, Priya Gaba, Akalewold Gebremeskel, Samantha Green, Gilbert Habonimana, Nicole Kozloff, Victoire Kpade, Pierre Lauzon, Andrew Mclellan, Van Nguyen, Anita Palepu, Nicole Pinto, Asia Rehman, Kim Van Herk, Jean Wang, Mackenzie Wilson, Vanessa Ymele Leki. Finally, the authors would like to thank Glenna Jenkins for her editorial input.

## Author Contributions

**Conceptualization:** David Ponka, Claire Kendall, Vicky Stergiopoulos, Peter Tugwell, Kevin Pottie.

**Data curation:** David Ponka, Eric Agbata, Claire Kendall, Vicky Stergiopoulos, Oreen Mendonca, Olivia Magwood, Ammar Saad, Bonnie Larson, Annie Huiru Sun, Neil Arya, Kednapa Thavorn, Anne Andermann, Peter Tugwell, Kevin Pottie.

**Formal analysis:** Eric Agbata, Oreen Mendonca, Olivia Magwood, Ammar Saad, Annie Huiru Sun, Neil Arya, Kednapa Thavorn.

**Funding acquisition:** David Ponka, Claire Kendall, Vicky Stergiopoulos, Peter Tugwell, Kevin Pottie.

**Methodology:** David Ponka, Claire Kendall, Vicky Stergiopoulos, Peter Tugwell, Kevin Pottie.

**Supervision:** Kevin Pottie.

**Writing – original draft:** David Ponka, Eric Agbata, Oreen Mendonca, Olivia Magwood, Ammar Saad, Bonnie Larson, Neil Arya, Kednapa Thavorn.

**Writing – review & editing:** David Ponka, Eric Agbata, Claire Kendall, Vicky Stergiopoulos, Oreen Mendonca, Olivia Magwood, Ammar Saad, Bonnie Larson, Annie Huiru Sun, Neil Arya, Terry Hannigan, Kednapa Thavorn, Anne Andermann, Peter Tugwell, Kevin Pottie.

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
