## [Decision Letter · Decision Letter 0]

28 Nov 2019

PONE-D-19-19380

The Effectiveness of Case-Management Interventions for the Homeless, Vulnerably
Housed and Persons with Lived Experience: A Systematic Review and Meta-Analysis

PLOS ONE

Dear Dr. Pottie,

Thank you for submitting your manuscript to PLOS ONE. After careful consideration, we
feel that it has merit but does not fully meet PLOS ONE’s publication criteria as it
currently stands. Therefore, we invite you to submit a revised version of the
manuscript that addresses the points raised during the review process.

The manuscript was reviewed by three reviewers. Although all Reviewers appreciated
the importance of the subject and the good style of writing, they do raise the
relevant methodological shortcomings of the manuscript that need to be thoroughly
reviewed. In particular, Reviewers raise doubts about the correctness of the
statistical analyses with reference to the meta-analysis. The Reviewers note that
Discussions also need to be reviewed. Therefore, I suggest that the Authors proceed
to address all the Reviewers’ comments to make the manuscript suitable for
publication.

We would appreciate receiving your revised manuscript by Jan 12 2020 11:59PM. When
you are ready to submit your revision, log on to https://www.editorialmanager.com/pone/ and select the 'Submissions
Needing Revision' folder to locate your manuscript file.

If you would like to make changes to your financial disclosure, please include your
updated statement in your cover letter.

To enhance the reproducibility of your results, we recommend that if applicable you
deposit your laboratory protocols in protocols.io, where a protocol can be assigned
its own identifier (DOI) such that it can be cited independently in the future. For
instructions see: http://journals.plos.org/plosone/s/submission-guidelines#loc-laboratory-protocols

We look forward to receiving your revised manuscript.

Kind regards,

Stefano Federici, Ph.D.

Academic Editor

PLOS ONE

Journal Requirements:

1. Please upload a copy of Figures 3 &4, to which you refer in your text on page
36. If the figure is no longer to be included as part of the submission please
remove all reference to it within the text.

Additional Editor Comments (if provided):

The manuscript was reviewed by three reviewers. Although all Reviewers appreciated
the importance of the subject and the good style of writing, they do raise the
relevant methodological shortcomings of the manuscript that need to be thoroughly
reviewed. In particular, Reviewers raise doubts about the correctness of the
statistical analyses with reference to the meta-analysis. The Reviewers note that
Discussions also need to be reviewed. Therefore, I suggest that the Authors proceed
to address all the Reviewers’ comments to make the manuscript suitable for
publication.

Reviewers' comments:

Reviewer's Responses to Questions

**Comments to the Author**

1. Is the manuscript technically sound, and do the data support the conclusions?

Reviewer #1: Yes

Reviewer #2: Partly

Reviewer #3: Partly

2. Has the statistical analysis been performed
appropriately and rigorously? 

Reviewer #1: Yes

Reviewer #2: Yes

Reviewer #3: No

3. Have the authors made all data underlying the
findings in their manuscript fully available?

Reviewer #1: Yes

Reviewer #2: Yes

Reviewer #3: Yes

4. Is the manuscript presented in an intelligible
fashion and written in standard English?

Reviewer #1: Yes

Reviewer #2: Yes

Reviewer #3: Yes

5. Review Comments to the Author

Reviewer #1: This is a review article on case management interventions for persons
who are homeless or otherwise unstably housed, focused on ACT, ICM, and CTI. The
effects of these interventions on health and social outcomes are discussed, as well
as the quality of the evidence surrounding each intervention. These interventions
are critical for improving care for a very vulnerable population and a systematic
evaluation of the evidence surrounding these practices is valuable. The article is
well-written. Some specific comments are offered below.

Abstract

• The conclusions of the abstract reflect the need to balance fidelity to an
intervention’s components (used to achieve outcomes shown in research settings) and
adaptation to meet the real-world context of under-resourced settings. To that end,
might be good to add a sentence to the background of the abstract highlighting the
clinical relevance of this review, i.e., what is the utility of studying these case
management approaches with regards to real world care.

• There seems to be a comparison between mainstream case management and three more
intensive CM models, but mainstream CM is not mentioned in the abstract

Introduction

• The first sentence references structural challenges (of which there are many) -
however, the second sentence describes individual level factors that are barriers to
care, as opposed to structural challenges.

• Table 1

o SCM - is this limited to persons engaging in primary care? I think that routine
case management happens for many homeless individuals who receive social services
but who do not receive health care services in primary care settings. A concern for
throughout the manuscript is that SCM is very challenging to define and very diverse
across settings and studies.

o CTI - generally this model is not just with any transition, but the transition
between an institutional setting to community living

• Interested to hear more about how the Delphi consensus panel helped prioritize
interventions of interest - were SCM, ICM, CTI, ACT selected because of this panel?
A few sentences about this in the methods as opposed to the intro might be
helpful.

Results

• The Hurlbu study (SCM) is a HUD-VASH study, which is a PSH program (I thought these
were excluded to be part of a separate review)

• Clark et al (2003) - the comprehensive housing program sounds very much like
PSH

• The findings on ACT - findings on hospitalization were mostly positive, which I
think is consistent with the literature. How does this work juxtaposed with the
assertion that benefits of ACT on mental health outcomes were minimal -
hospitalization is generally a proxy for mental health outcomes.

• Cost effectiveness: how could total cost incurred by SCM clients be greater than
total cost for ACT clients?

Discussion

• Appreciate the recognition at the top of the discussion that the interventions,
populations, and outcomes were heterogeneous, which makes it hard to analyze the
data - it would be good to have this sort of statement up front at the end of the
introduction statement

• A fundamental question is if SCM is good for a different subgroup of homeless
people than the more intensive interventions studied. Or, is SCM just too low
intensity for homeless people, but acceptable for a less vulnerable population? The
article basically says that more intensive case management is needed for everyone in
this population - I’m not sure that is true, as there has got to be some match
between clinical acuity and the acuity of the CM intervention needed; SCM is also
such a broad and vague term encompassing many different things in studies and in
real-world settings

• There is a statement in the discussion that persons with lived experience were
integrated on the research team - not clear how this took place or in what capacity
they were involved in these analyses

Reviewer #2: Thank you for the opportunity to review this interesting manuscript.
Since I am not an expert in the topic, my comments will focus on the systematic
review and meta-analysis methodology and reporting.

Major Concern:

The results of this systematic review and meta-analysis should focus more on the
meta-analytic findings. Nearly all of the results presented resort to counting
p-values or describing the findings from specific studies. The beauty of
meta-analysis is to move beyond p-counting so that you can show the overall effect
of a type of intervention. Of the 56 studies in this review, I’m only seeing 3
studies meta-analyzed, for only two outcomes. No rationale is provided for why a
meta-analysis was not performed on any other outcomes.

Even the narrative review results are rudimentary, with very little synthesis
reported.

Additional comments:

1. The protocol has not been published, so reviewers only have the limited methods
section to use to evaluate the review methods. While the methods appear to be
rigorous, more detail would be helpful, particularly in the data analysis section.
Also, the author guidelines for PLOS One state that systematic reviews without
published protocols should include the protocol in the supplementary material.

2. Was the protocol registered with a systematic review registry, such as PROSPERO? I
did not see a PROSPERO registration number mentioned. PLOS One submission guidelines
require the protocol registry number to be included in the abstract.

3. Please address in the methods how you managed studies that had multiple
interventions using the same control group in your meta-analyses.

4. The PRISMA flow diagram should include a summary of the reasons for exclusion for
the 214 studies excluded at the full-text review stage in the original search. Don’t
just make readers go to the supplementary files and then expect the reader to count
the reasons themselves. While I admire the level of detail you’re including in your
tables and supplementary files, the whole point of a review is to summarize and
synthesize for readers.

5. Table 9 needs a legend that defines the abbreviations for the intervention
types.

6. On p. 35, line 189, the word “trivial” may not be the best word choice, as it can
imply a value judgement. Perhaps something like “equivocal” would be better?

7. Are Figures 1 and 2 data from just one study? This is not clear from the figures
alone, as there is no study cited, and forest plots typically imply meta-analytic
results across studies.

8. The figures on page 36 are not labeled.

Reviewer #3: Overall, this is a well-written analysis of case management
interventions of homeless or vulnerably-housed individuals. This is not my field of
study, so I cannot comment on the relevance of this review to this field or the
quality of the qualitative analysis portion, other than to say that it was thorough.
However, I have some concerns regarding the meta-analysis. It seems improper to use
a random-effects meta-analysis in a meta-analysis including only 2 or 3 studies. My
suspicion is that, in practice, your conclusion that ICM reduces number of days
homeless will be robust to either approach: fixed effects vs. random effects, but it
would be encouraging to know that the random effects approach has not
under-estimated the confidence intervals on the pooled effect. I would recommend
repeating the analyses from the figures on page 36 using a fixed effects approach
(basically an inverse-variance weighted average of the studies) and indicate whether
the results were similar or not to the random effects results.

Minor comments:

Tables 4-8 should go into a supplemental material file. This is a thorough summary of
all of these studies and way too much information for the text of the
manuscript.

Figures 1 and 2: Are these from a single study? If so, I don't understand why it's
necessary to present these findings that have been presented elsewhere. I would
recommend dropping these figures and succinctly summarizing the results in the
text.

6. PLOS authors have the option to publish the peer
review history of their article (what does this mean?). If published, this will
include your full peer review and any attached files.

If you choose “no”, your identity will remain anonymous but your review may still be
made public.

**Do you want your identity to be public for this peer review?** For
information about this choice, including consent withdrawal, please see our
Privacy Policy.

Reviewer #1: Yes: Sonya Gabrielian

Reviewer #2: No

Reviewer #3: No

---

## [Author Response · Author response to Decision Letter 0]

30 Dec 2019

Kevin Pottie, MD CCFP MCISc FCFP

85 Primrose Ave, room 307, Ottawa, ON

kpottie@uottawa.ca

Re: The Effectiveness of Case-Management Interventions for the Homeless, Vulnerably
Housed and Persons with Lived Experience: A Systematic Review and Meta-Analysis
(reference number PONE-D-19-19380)

Response to editors and peer-reviewers

Editorial comments

Please ensure that your manuscript meets PLOS ONE's style requirements, including
those for file naming

Response: Thank you. We have reviewed PLOS ONE’s style requirements and have edited
our manuscript as necessary.

Please upload a copy of Figures 3 &4, to which you refer in your text on page 36.
If the figure is no longer to be included as part of the submission please remove
all reference to it within the text.

Response: Figures 3 and 4 are the forest plots representing our pooled meta-analyses.
We have uploaded them separately as per your request.

Peer reviewers’ comments

Reviewer #1

This is a review article on case management interventions for persons who are
homeless or otherwise unstably housed, focused on ACT, ICM, and CTI. The effects of
these interventions on health and social outcomes are discussed, as well as the
quality of the evidence surrounding each intervention. These interventions are
critical for improving care for a very vulnerable population and a systematic
evaluation of the evidence surrounding these practices is valuable. The article is
well-written. Some specific comments are offered below.

Response: Thank you for taking the time to review our manuscript and highlight the
importance of such a comprehensive review of the literature on the effectiveness and
cost effectiveness of case management interventions among homeless and vulnerably
housed populations. We have addressed your comments and feedback. Kindly see our
responses below.

Abstract

• The conclusions of the abstract reflect the need to balance fidelity to an
intervention’s components (used to achieve outcomes shown in research settings) and
adaptation to meet the real-world context of under-resourced settings. To that end,
might be good to add a sentence to the background of the abstract highlighting the
clinical relevance of this review, i.e., what is the utility of studying these case
management approaches with regards to real world care.

Response: Thank you for this comment. The definitions of case management models in
the very heterogeneous literature makes this addition very relevant. This also links
to your comment below about SCM which we address further. In fact, our conclusions
reflect the need to better understand the link between intensity and effect,
especially in complex and variable primary care settings.

• There seems to be a comparison between mainstream case management and three more
intensive CM models, but mainstream CM is not mentioned in the abstract

Response: Thank you for highlighting this discrepancy. We have added to our abstract
findings of mainstream case management

Introduction

• The first sentence references structural challenges (of which there are many) -
however, the second sentence describes individual level factors that are barriers to
care, as opposed to structural challenges.

Response: Thank you, we have amended these sentences to improve clarity. 

• Table 1

o SCM - is this limited to persons engaging in primary care? I think that routine
case management happens for many homeless individuals who receive social services
but who do not receive health care services in primary care settings. A concern for
throughout the manuscript is that SCM is very challenging to define and very diverse
across settings and studies.

Response: Thank you for this comment. We agree that SCM is very heterogeneous. We
have added additional remarks to that effect on our introduction. SCM is also not
limited to persons engaging in primary care, although it is common. We have revised
Table 1 to reflect that SCM may be delivered to persons with complex care needs. 

o CTI - generally this model is not just with any transition, but the transition
between an institutional setting to community living

Response: We sought evidence on critical time interventions that aimed to help
individuals transition from a state of precarious housing and into more stable
accommodation. As a result of our search, we found evidence on three CTI trials that
assist individuals transition from shelters to more stable housing as well as one
trial devising a transition plan after discharge from the hospital. Thank you for
you comment. We will make this more clear in Table 1 and the results section.

• Interested to hear more about how the Delphi consensus panel helped prioritize
interventions of interest - were SCM, ICM, CTI, ACT selected because of this panel?
A few sentences about this in the methods as opposed to the intro might be
helpful.

Response: Thank you for this comment. We have provided additional details in the
methods (see section “Selection of priority interventions”) regarding the Delphi
process and the rationale for choosing these models of case management. 

Results

• The Hurlbut study (SCM) is a HUD-VASH study, which is a PSH program (I thought
these were excluded to be part of a separate review)

Response: Thank you. Even though the Hurlburt study is part of the HUD-VASH project.
The findings of the two publications reported on the added benefits of providing
case management to participants regardless of their housing arrangements. We sought
evidence on every study that reported on the benefits (or harms, if found) of case
management interventions, and thus we have included these publications for that
purpose.

• Clark et al (2003) - the comprehensive housing program sounds very much like
PSH

Response: You are correct. The comprehensive housing program is very much a
replication of PSH. However, participants allocated to the other arm of this trial
are provided with case management services only. We included this study in our
analysis to assess whether case management only was superior or equivalent to PSH in
improving our outcomes of interest.

• The findings on ACT - findings on hospitalization were mostly positive, which I
think is consistent with the literature. How does this work juxtaposed with the
assertion that benefits of ACT on mental health outcomes were minimal -
hospitalization is generally a proxy for mental health outcomes.

Response: We agree and have rephrase the sentence to reflect the reasonably positive
effects of ACT interventions in reducing in psychiatric symptoms across studies. 

• Cost effectiveness: how could total cost incurred by SCM clients be greater than
total cost for ACT clients?

Response: Thank you for your comments. The total cost of SCM was higher than ACT in
the case when the studies included all cost components incurred to society (termed
the ‘societal perspective’). Specifically, the slightly higher cost of SCM was due
to the fact that SCM clients had nominally more frequent visits to outpatient health
care and other community services, more arrest episodes, and incurred higher family
time costs compared to ACT clients. We have clarified this point in the cost and
cost-effectiveness section.

Discussion

• Appreciate the recognition at the top of the discussion that the interventions,
populations, and outcomes were heterogeneous, which makes it hard to analyze the
data - it would be good to have this sort of statement up front at the end of the
introduction statement

Response: We agree, and have provided two sentences at the end of the introduction to
highlight the heterogeneity of the interventions at hand.

• A fundamental question is if SCM is good for a different subgroup of homeless
people than the more intensive interventions studied. Or, is SCM just too low
intensity for homeless people, but acceptable for a less vulnerable population? The
article basically says that more intensive case management is needed for everyone in
this population - I’m not sure that is true, as there has got to be some match
between clinical acuity and the acuity of the CM intervention needed; SCM is also
such a broad and vague term encompassing many different things in studies and in
real-world settings

Response: Thank you very much for these thoughtful questions. We agree that the
definition of SCM is challenging and thus decided to take a broad approach. We
attempted to distinguish the differences in CM in Table 1. However, comparing SCM to
more intensive models was not the focus of the review, and indeed such a comparison
would likely be based on comparable populations - per Table 1, the more intensive
approaches in fact aimed at specific subsets of homeless popullations. Because of
all these factors, not to mention that issues of fidelity that you raise, we
tempered the strength of our conclusions.

• There is a statement in the discussion that persons with lived experience were
integrated on the research team - not clear how this took place or in what capacity
they were involved in these analyses

Response: Thank you for highlighting the shortage in describing the role of our
people with lived experience of homelessness in this project. We have provided more
information regarding this issue in the methods section.

Reviewer #2: Thank you for the opportunity to review this interesting manuscript.
Since I am not an expert in the topic, my comments will focus on the systematic
review and meta-analysis methodology and reporting.

Response: Thank you for reviewing our work and providing feedback on our methods. An
important strength of our review is the rigorous methodology we have used in
screening, data collection and management, data analysis, and reporting processes.
We have addressed all your comments. Kindly find our responses below. 

Major Concern:

The results of this systematic review and meta-analysis should focus more on the
meta-analytic findings. Nearly all of the results presented resort to counting
p-values or describing the findings from specific studies. The beauty of
meta-analysis is to move beyond p-counting so that you can show the overall effect
of a type of intervention. Of the 56 studies in this review, I’m only seeing 3
studies meta-analyzed, for only two outcomes. No rationale is provided for why a
meta-analysis was not performed on any other outcomes.

Response: Thanks, meta-analysis is one of the most robust statistical methods to
synthesize outcome data to provide a quantitative estimate and we used this method
whenever outcome data permitted. We attempted to meta-analyze all outcomes, but
heterogeneity in interventions, outcomes and time-points precluded this for the
majority of outcomes. We have clarified this in the methods section.

Even the narrative review results are rudimentary, with very little synthesis
reported.

Response: Thank you. It appears we did not clearly describe our synthesis. Given that
the majority of our synthesis is narrative, we have referred to the SWiM (Synthesis
Without Meta-Analysis) reporting guidelines (previously named “Improving the Conduct
and reporting of Narrative Synthesis of Quantitative data (ICONS-Quant)”), which is
presently under consideration for publication (must be kept confidential). The
ICONS-Quant items are intended to complement PRISMA. ICONS-Quant relates to the
methods and reporting of narrative synthesis, while PRISMA relates to the entire
systematic review process. We have modified our paper to reflect these reporting
items.

Additional comments:

1. The protocol has not been published, so reviewers only have the limited methods
section to use to evaluate the review methods. While the methods appear to be
rigorous, more detail would be helpful, particularly in the data analysis section.
Also, the author guidelines for PLOS One state that systematic reviews without
published protocols should include the protocol in the supplementary material.

Response: Thank you for this comment. Our protocol was published during the
peer-review period. We have updated our reference list and provide a DOI to the
open-access publication for reviewers to consider. Please see: https://doi.org/10.1002/cl2.1048

2. Was the protocol registered with a systematic review registry, such as PROSPERO? I
did not see a PROSPERO registration number mentioned. PLOS One submission guidelines
require the protocol registry number to be included in the abstract.

Response: Thank you for this question. We registered our title (2018) and protocol
(2019) with the Campbell Collaboration, which is considered a systematic review
registry. We have made this more explicit in our manuscript under “Methods”. We have
updated our reference list and provide a DOI to the open-access publication for
reviewers to consider.

3. Please address in the methods how you managed studies that had multiple
interventions using the same control group in your meta-analyses.

Response: Thank you for this comment. We have added the following to our methods
section to clarify this point: To prevent double-counting of outcomes, individual
records were carefully screened to identify unique trial studies. Each study was
then evaluated for potential overlap using study design, enrollment and data
collection dates, authors and their associated affiliations and the reported
selection and eligibility criteria in the studies to inform the assessment. Studies
deemed to be at risk for double-counting were discussed by the research team and
decisions for inclusion in meta-analysis (and any additional analyses) were
made.

4. The PRISMA flow diagram should include a summary of the reasons for exclusion for
the 214 studies excluded at the full-text review stage in the original search. Don’t
just make readers go to the supplementary files and then expect the reader to count
the reasons themselves. While I admire the level of detail you’re including in your
tables and supplementary files, the whole point of a review is to summarize and
synthesize for readers.

Response: Thank you for this comment. We have updates this figure with the following
information:

Excluded studies n=214

Reasons:

Wrong study design n=51

Irrelevant outcomes n=8

Wrong population n=25

Wrong intervention n=117

Wrong publication type n=117 10

Could not be retrieved n=3

5. Table 9 needs a legend that defines the abbreviations for the intervention
types.

Response: Thank you, we have added a legend to this table which defines the
abbreviations. 

6. On p. 35, line 189, the word “trivial” may not be the best word choice, as it can
imply a value judgement. Perhaps something like “equivocal” would be better?

Response: Thank you for this suggestion, we have made the appropriate changes. 

7. Are Figures 1 and 2 data from just one study? This is not clear from the figures
alone, as there is no study cited, and forest plots typically imply meta-analytic
results across studies.

Response: Thank you, these figures have been removed.

8. The figures on page 36 are not labeled.

Response: Thank you, we have received feedback against using these forest plots in
our results section as they come from the same study, and therefore, we have decided
to replace them with a descriptive synthesis of evidence from these studies.

Reviewer #3: Overall, this is a well-written analysis of case management
interventions of homeless or vulnerably-housed individuals. This is not my field of
study, so I cannot comment on the relevance of this review to this field or the
quality of the qualitative analysis portion, other than to say that it was thorough.
However, I have some concerns regarding the meta-analysis. It seems improper to use
a random-effects meta-analysis in a meta-analysis including only 2 or 3 studies. My
suspicion is that, in practice, your conclusion that ICM reduces number of days
homeless will be robust to either approach: fixed effects vs. random effects, but it
would be encouraging to know that the random effects approach has not
under-estimated the confidence intervals on the pooled effect. I would recommend
repeating the analyses from the figures on page 36 using a fixed effects approach
(basically an inverse-variance weighted average of the studies) and indicate whether
the results were similar or not to the random effects results.

Response: Thank you for this comment. As per your suggestion, we have rerun the
analysis using both a fixed effects model and found that our conclusions were
unchanged:

FIXED EFFECT MODEL:

RANDOM EFFECT MODEL:

However, we stand behind our original decision to publish the meta-analysis using a
random effects model due to its consideration of heterogeneity. Under any
interpretation, a fixed-effect meta-analysis ignores heterogeneity. In the
meta-analyzed studies of our review, we have conceptual heterogeneity in populations
(e.g. youth [Grace 2014] vs. mentally ill adults with children [Toro 1997] vs.
chronic inebriated adults [Cox 1998]) and interventions ( time-limited ICM [Grace
2014] vs. ICM with job training [Toro 1997] vs long-term ICM [Cox 1998]). Given this
conceptual heterogeneity, we could not assume that the true effect of intervention
(in both magnitude and direction) is the same value in every study (i.e. fixed
across studies). Instead of assuming that the intervention effects are the same, we
assume that they follow (usually) a normal distribution. The assumption implies that
the observed differences among study results are due to a combination of the play of
chance and some genuine variation in the intervention effects.

Minor comments:

Tables 4-8 should go into a supplemental material file. This is a thorough summary of
all of these studies and way too much information for the text of the
manuscript.

Response: Thank you for this suggestion, we have moved these tables to supplemental
material (S4).

Figures 1 and 2: Are these from a single study? If so, I don't understand why it's
necessary to present these findings that have been presented elsewhere. I would
recommend dropping these figures and succinctly summarizing the results in the
text.

Response: Thank you for this suggestion, we have removed these figures.

1 - Ponka et al. - Response to reviewers Nov
2019.docx
---

## [Decision Letter · Decision Letter 1]

28 Jan 2020

PONE-D-19-19380R1

The Effectiveness of Case-Management Interventions for the Homeless, Vulnerably
Housed and Persons with Lived Experience: A Systematic Review and Meta-Analysis

PLOS ONE

Dear Dr. Pottie,

Thank you for submitting your manuscript to PLOS ONE. After careful consideration, we
feel that it has merit but does not fully meet PLOS ONE’s publication criteria as it
currently stands. Therefore, we invite you to submit a revised version of the
manuscript that addresses the points raised during the review process.

In this second round of review, I
invite the author to take more careful consideration of the valuable notes of
the Reviewers, following their suggestions and responding more fully to the
objections raised.

We would appreciate receiving your revised manuscript by Mar 13 2020 11:59PM. When
you are ready to submit your revision, log on to https://www.editorialmanager.com/pone/ and select the 'Submissions
Needing Revision' folder to locate your manuscript file.

If you would like to make changes to your financial disclosure, please include your
updated statement in your cover letter.

To enhance the reproducibility of your results, we recommend that if applicable you
deposit your laboratory protocols in protocols.io, where a protocol can be assigned
its own identifier (DOI) such that it can be cited independently in the future. For
instructions see: http://journals.plos.org/plosone/s/submission-guidelines#loc-laboratory-protocols

We look forward to receiving your revised manuscript.

Kind regards,

Stefano Federici, Ph.D.

Academic Editor

PLOS ONE

Additional Editor Comments (if provided):

In this second round of review, I invite the author to take more careful
consideration of the valuable notes of the Reviewers, following their suggestions
and responding more fully to the objections raised.

Reviewers' comments:

Reviewer's Responses to Questions

**Comments to the Author**

1. If the authors have adequately addressed your comments raised in a previous round
of review and you feel that this manuscript is now acceptable for publication, you
may indicate that here to bypass the “Comments to the Author” section, enter your
conflict of interest statement in the “Confidential to Editor” section, and submit
your "Accept" recommendation.

Reviewer #1: All comments have been addressed

Reviewer #2: (No Response)

Reviewer #3: (No Response)

2. Is the manuscript technically sound, and do the data
support the conclusions?

Reviewer #1: Yes

Reviewer #2: Yes

Reviewer #3: Yes

3. Has the statistical analysis been performed
appropriately and rigorously? 

Reviewer #1: Yes

Reviewer #2: Yes

Reviewer #3: Yes

4. Have the authors made all data underlying the
findings in their manuscript fully available?

Reviewer #1: Yes

Reviewer #2: Yes

Reviewer #3: Yes

5. Is the manuscript presented in an intelligible
fashion and written in standard English?

Reviewer #1: Yes

Reviewer #2: Yes

Reviewer #3: Yes

6. Review Comments to the Author

Reviewer #1: The authors sufficiently addressed the concerns I mentioned in the first
round of reviews. The only thing missing from the discussion section at this point
from my perspective is a more nuanced reflection that homeless people have different
levels of acuity and thus have different levels of need for case management
intensity. That is, ACT isn't for everyone, ICM isn't for everyone, these are case
management models needed for certain people with certain functional limitations and
there should be some effort to match needs to services.

Reviewer #2: I’d like to thank the authors for their response to the reviewer
comments. I understand how challenging it is to do a project of this scope.

The authors’ response to the critique about registering the systematic review is
somewhat misleading. Publishing a review protocol is not the same as registering the
protocol. Submitting the title to the Campbell Collaboration is not the same as
listing the planned review in a publicly accessible registry prior to conducting the
review. Further, the authors’ wording on p. 5 (lines 86-87) may mislead readers to
think that this review was done under the auspices or sponsorship of the Campbell
Collaboration (“…published by the Campbell Collaboration”), but my reading of the
review protocol is that this was not the case. It might be simpler and more accurate
to state that the review protocol was not registered, but was published in 2019 (and
then cite the protocol paper).

I still think that the authors’ narrative synthesis of results is rather rudimentary,
but I understand that it can be difficult to write a narrative synthesis without
falling into the trap of describing things study-by-study.

I’d like to put in a note of support for the authors’ response to the first comment
from Reviewer 3. I agree with the authors that a random effects model is the
appropriate approach for meta-analyzing these types of studies, for the very reasons
they give in their rationale.

Ultimately, however, this manuscript is kind of a bait-and-switch. The title tempts a
reader with a meta-analysis, but then out of the 56 studies included, only 3 studies
were used for two (related) outcomes for only one of the types of case management. I
have a genuine concern that the narrative review findings from this manuscript will
be cited as though they are based on a meta-analysis when in fact they aren’t. While
I don’t want to take away from the authors the work they did on the small
meta-analysis in this paper, I also think that it might be more honest to remove
“and Meta-analysis” from the title, since the overwhelming majority of this review’s
findings are not based on meta-analysis.

Reviewer #3: Regarding the fixed effects vs. random effects issue, the problem isn't
that you don't have heterogeneity. The problem is that it's hard to quantify the
heterogeneity with only 2-3 studies. The fact that the fixed effects analysis has
qualitatively similar results should be briefly mentioned in the text and the
results should be added to supplemental information.

7. PLOS authors have the option to publish the peer
review history of their article (what does this mean?). If published, this will
include your full peer review and any attached files.

If you choose “no”, your identity will remain anonymous but your review may still be
made public.

**Do you want your identity to be public for this peer review?** For
information about this choice, including consent withdrawal, please see our
Privacy Policy.

Reviewer #1: Yes: Sonya Gabrielian

Reviewer #2: No

Reviewer #3: No

---

## [Author Response · Author response to Decision Letter 1]

18 Feb 2020

Ponka et al. Case Management Review (Revision 2) Response to Reviewers Feb 2020

Reviewer #1: The authors sufficiently addressed the concerns I mentioned in the first
round of reviews. The only thing missing from the discussion section at this point
from my perspective is a more nuanced reflection that homeless people have different
levels of acuity and thus have different levels of need for case management
intensity. That is, ACT isn't for everyone, ICM isn't for everyone, these are case
management models needed for certain people with certain functional limitations and
there should be some effort to match needs to services.

Response: Thank you for reviewing our manuscript and for your helpful feedback. We
agree that not all models of case management are appropriate for every individual,
and that the services provided should be matched to client needs. We have edited our
discussion section and have included the following: “However, it is likely that a
dose-response relationship may explain some of our findings, and that as higher
intensity interventions such as ACT and ICM are more precisely defined, there may be
greater attention to fidelity in their implementation (19). Alternatively, it is
possible that lower intensity models work predominantly for homeless populations
with less acute issues (or for those that are precariously housed), and this would
suggest the importance of matching the intensity of the intervention with the acuity
of need.” 

Reviewer #2: I’d like to thank the authors for their response to the reviewer
comments. I understand how challenging it is to do a project of this scope.

Response: Thank you for your support. 

The authors’ response to the critique about registering the systematic review is
somewhat misleading. Publishing a review protocol is not the same as registering the
protocol. Submitting the title to the Campbell Collaboration is not the same as
listing the planned review in a publicly accessible registry prior to conducting the
review. Further, the authors’ wording on p. 5 (lines 86-87) may mislead readers to
think that this review was done under the auspices or sponsorship of the Campbell
Collaboration (“…published by the Campbell Collaboration”), but my reading of the
review protocol is that this was not the case. It might be simpler and more accurate
to state that the review protocol was not registered, but was published in 2019 (and
then cite the protocol paper).

Response: Thank you for this feedback. You are correct; the protocol for this review
was not registered in a publicly available registry and we agree that the original
reporting of this in our manuscript may be misleading. We have edited this paragraph
of the methods section to be explicit about this and provide an explicit statement
for our reader: “We conducted a systematic review according to a published
peer-reviewed protocol (29). The protocol was not registered in an open-access
registry (e.g. PROSPERO) prior to publication”. Thank you for bringing this to our
attention, this is an important lesson learned for the future. 

I still think that the authors’ narrative synthesis of results is rather rudimentary,
but I understand that it can be difficult to write a narrative synthesis without
falling into the trap of describing things study-by-study.

Response: Thank you for this comment. Despite frequent use and longstanding concerns
about the validity of narrative synthesis, there has been almost no methodological
development to promote clearer methods for narrative synthesis, despite its use in
nearly half of health related systematic reviews. We have attempted to tabulate
meaningful results (Table 4), a data presentation method suggested by the SWiM
reporting guidelines, but we do recognize the limited richness of this synthesis. We
appreciate your understanding. 

I’d like to put in a note of support for the authors’ response to the first comment
from Reviewer 3. I agree with the authors that a random effects model is the
appropriate approach for meta-analyzing these types of studies, for the very reasons
they give in their rationale.

Response: Thank you very much for this support. We have worked hard to achieve
methodological rigor in our review and are confident in this decision.

Ultimately, however, this manuscript is kind of a bait-and-switch. The title tempts a
reader with a meta-analysis, but then out of the 56 studies included, only 3 studies
were used for two (related) outcomes for only one of the types of case management. I
have a genuine concern that the narrative review findings from this manuscript will
be cited as though they are based on a meta-analysis when in fact they aren’t. While
I don’t want to take away from the authors the work they did on the small
meta-analysis in this paper, I also think that it might be more honest to remove
“and Meta-analysis” from the title, since the overwhelming majority of this review’s
findings are not based on meta-analysis.

Response: This is an excellent point. We agree and have removed “and meta-analysis”
from our title.

Reviewer #3: Regarding the fixed effects vs. random effects issue, the problem isn't
that you don't have heterogeneity. The problem is that it's hard to quantify the
heterogeneity with only 2-3 studies. The fact that the fixed effects analysis has
qualitatively similar results should be briefly mentioned in the text and the
results should be added to supplemental information.

Response: Thank you, this is a good idea. We have added a sentence to our methods
section stating that we ran these parallel analyses, and have written in the results
section that findings were unchanged (i.e. not dependent on fixed vs. random effects
models). We have included these results in Appendix S7. We appreciate your
constructive feedback.

2 - Ponka et al. - Response to Reviewers.docx
---

## [Decision Letter · Decision Letter 2]

12 Mar 2020

The Effectiveness of Case-Management Interventions for the Homeless, Vulnerably
Housed and Persons with Lived Experience: A Systematic Review

PONE-D-19-19380R2

Dear Dr. Pottie,

We are pleased to inform you that your manuscript has been judged scientifically
suitable for publication and will be formally accepted for publication once it
complies with all outstanding technical requirements.

With kind regards,

Stefano Federici, Ph.D.

Academic Editor

PLOS ONE

Additional Editor Comments (optional):

Reviewers' comments:

Reviewer's Responses to Questions

**Comments to the Author**

1. If the authors have adequately addressed your comments raised in a previous round
of review and you feel that this manuscript is now acceptable for publication, you
may indicate that here to bypass the “Comments to the Author” section, enter your
conflict of interest statement in the “Confidential to Editor” section, and submit
your "Accept" recommendation.

Reviewer #1: All comments have been addressed

Reviewer #2: All comments have been addressed

Reviewer #3: All comments have been addressed

2. Is the manuscript technically sound, and do the data
support the conclusions?

Reviewer #1: Yes

Reviewer #2: Yes

Reviewer #3: (No Response)

3. Has the statistical analysis been performed
appropriately and rigorously? 

Reviewer #1: Yes

Reviewer #2: Yes

Reviewer #3: (No Response)

4. Have the authors made all data underlying the
findings in their manuscript fully available?

Reviewer #1: Yes

Reviewer #2: Yes

Reviewer #3: (No Response)

5. Is the manuscript presented in an intelligible
fashion and written in standard English?

Reviewer #1: Yes

Reviewer #2: Yes

Reviewer #3: (No Response)

6. Review Comments to the Author

Reviewer #1: (No Response)

Reviewer #2: (No Response)

Reviewer #3: (No Response)

7. PLOS authors have the option to publish the peer
review history of their article (what does this mean?). If published, this will
include your full peer review and any attached files.

If you choose “no”, your identity will remain anonymous but your review may still be
made public.

**Do you want your identity to be public for this peer review?** For
information about this choice, including consent withdrawal, please see our
Privacy Policy.

Reviewer #1: No

Reviewer #2: No

Reviewer #3: No

---

## [Editor Report · Acceptance letter]

27 Mar 2020

PONE-D-19-19380R2 

The Effectiveness of Case Management Interventions for the Homeless, Vulnerably
Housed and Persons with Lived Experience: A Systematic Review. 

Dear Dr. Pottie:

I am pleased to inform you that your manuscript has been deemed suitable for
publication in PLOS ONE. Congratulations! Your manuscript is now with our production
department. 

With kind regards,

on behalf of

Prof. Stefano Federici 

Academic Editor

PLOS ONE